# QuadEnhancer: Leveraging Quadratic Transformations to Enhance Deep Neural Networks

Qian Chen[1,2], Linxin Yang[2,3], Akang Wang[2,3,*], Xiaodong Luo[2,3], and Yin Zhang[3,*]

[1]School of Science and Engineering, The Chinese University of Hong Kong, Shenzhen, China
[2]Shenzhen International Center for Industrial and Applied Mathematics, Shenzhen Research Institute of Big Data, China
[3]School of Data Science, The Chinese University of Hong Kong, Shenzhen, China

## Abstract

The combination of linear transformations and non-linear activation functions forms the foundation of most modern deep neural networks, enabling them to approximate highly complex functions. This paper explores the introduction of quadratic transformations to further increase nonlinearity in neural networks, with the aim of enhancing the performance of existing architectures. To reduce parameter complexity and computational complexity, we propose a lightweight quadratic enhancer that uses low-rankness, weight sharing, and sparsification techniques. For a fixed architecture, the proposed approach introduces quadratic interactions between features at every layer, while only adding negligible amounts of additional model parameters and forward computations. We conduct a set of proof-of-concept experiments for the proposed method across three tasks: image classification, text classification, and fine-tuning large-language models. In all tasks, the proposed approach demonstrates clear and substantial performance gains.

## 1 Introduction

In modern deep learning, the majority of successful architectures are built on the combination of linear transformations followed by non-linear activation functions. This fundamental framework allows neural networks to learn complex mappings from input to output. The linear transformation serves to project the input data into a different space, while the non-linear activation function introduces the necessary complexity, enabling the network to model intricate, high-dimensional patterns.

The introduction of non-linearity has been a key factor in the success of neural networks, enabling them to approximate highly complex functions and capture intricate data patterns. This ability is what allows neural networks to tackle problems ranging from image classification to natural language processing. Over the years, significant advancements have been made in the architecture of neural networks, starting with MLPs for simple regression and classification tasks [38, 1], followed by CNNs for image data [22, 21, 12, 23], RNNs and LSTMs for sequential data [14, 42, 25, 24], and more recently, Transformer models [45] that dominate a wide range of fields, including natural language processing [35, 5, 34], computer vision [19, 6], and beyond [46, 18].

The success of these architectures highlights the importance of incorporating non-linearity to achieve the level of model expressiveness needed for complex real-world tasks. As a result, researchers have continually sought ways to introduce more advanced non-linear transformations within neural networks to enhance their capabilities. These explorations have generally followed three main

---

*Corresponding authors: Yin Zhang <yinzhang@cuhk.edu.cn >, Akang Wang <wangakang@sribd.cn>

directions: (i) employing more complex activation functions, (ii) designing non-linear network modules, and (iii) replacing linear operations with polynomial transformations.

Firstly, recent advancements in neural network activation functions have focused on introducing more complex forms of non-linearity to improve model expressiveness and tackle real-world tasks. For instance, Swish [37], which combines a sigmoid function with a linear term, offers a smoother, more flexible non-linearity, outperforming ReLU in deep networks. Further enhancements include Mish [31], which introduces a combination of tanh and softplus to provide even finer gradients, and GELU [13], a probabilistic function used in Transformer models, which incorporates multiple non-linear operations such as tanh and cubic terms to introduce smooth, complex non-linearity. Despite their success, all these activation functions focus on element-wise transformations, capturing non-linearity at the level of individual neurons, yet often fail to exploit the potential for interactions between neurons that could further enhance representational capacity.

Secondly, there have been attempts to design non-linear network modules that go beyond the simple application of activation functions. For example, the introduction of GRU [3] and LSTM [14] units in recurrent networks adds non-linearity by incorporating gates that regulate the flow of information over time. Additionally, the attention mechanism [45], introduces context-dependent weighting of neurons, allowing the network to focus on more relevant parts of the input. While these modular approaches have shown great success, they are often task-specific and depend on network architecture, making them less universally applicable.

Finally, replacing linear operations with polynomial transformations represents a more radical approach to non-linearity. This approach includes polynomial networks [4], which replace standard linear layers with polynomial operations and reduce the number of parameters needed for higher-order terms through tensor decomposition techniques. Recently, [28] extended second-order methods to convolutional neural networks. [7, 8] explored the advantages of quadratic transformations in terms of both representational power and training efficiency. More recently, [48, 47] combined quadratic methods with neural architecture search, further improving model performance. These methods have demonstrated both theoretically and experimentally that higher-order transformations can enhance a model's representational capacity, thereby boosting its overall performance. However, these approaches have typically been limited due to the substantial increase in parameters and computational cost associated with higher-order terms.

While methods involving more complex activation functions and specialized network modules have demonstrated significant success, polynomial transformations—despite their theoretical potential—have been underexplored in many real-world applications. The main challenge lies in the fact that higher-order terms require a considerable number of parameters, which can significantly increase model complexity. For instance, even with aggressive decomposition techniques, the work of [4] still requires $O(n^2)$ parameters in high-order terms. Nevertheless, as a complementary approach to existing methods, higher-order transformations offer the potential to further enhance expressiveness without conflicting with activation functions or specialized network modules. This motivates the investigation of how polynomial transformations can be integrated with standard non-linearities to improve model performance while maintaining control over computational complexity.

In this paper, we propose a novel enhancement to traditional neural network architectures by introducing a quadratic transformation at each linear layer. This quadratic enhancement introduces higher-order interactions between neurons through quadratic terms, while maintaining computational efficiency by reusing the linear activation outputs and sparsifying parameter matrix of the quadratic term. Our method significantly reduces the number of parameters and operations required for standard quadratic transformations, making it light-weight, easily applicable to modern architectures. The key contributions of this paper are as follows:

- We introduce a novel quadratic transformation technique that enhances non-linearity in neural networks by leveraging quadratic transformations to capture richer interactions between neurons.

- We present a sparsified version of the quadratic transformation that reduces the number of parameters, ensuring that the additional computational overhead remains minimal.

- We evaluate the effectiveness of our approach through extensive experiments on three tasks, including image classification, text classification, and fine-tuning of large language models (LLMs), showing substantial performance improvements over baseline models.

## 2 Preliminaries

### 2.1 Notation

Unless otherwise specified, scalars are denoted by normal font (e.g., $x$, $y$), vectors are denoted by bold lowercase letters (i.e., $\boldsymbol{x}$, $\boldsymbol{\lambda}$), and matrices are denoted by bold uppercase letters (e.g., $\boldsymbol{W}$, $\boldsymbol{P}$ and $\boldsymbol{\Lambda}$).

### 2.2 Standard linear transformation

A typical linear transformation converts an input signal $\boldsymbol{x} \in \mathbb{R}^n$ to a feature vector $\boldsymbol{y} \in \mathbb{R}^d$ by multiplying a weight matrix $\boldsymbol{W} \in \mathbb{R}^{d \times n}$ as

$$\boldsymbol{y} := \boldsymbol{W}\boldsymbol{x} + \boldsymbol{b}. \tag{1}$$

The transformed vector $\boldsymbol{y}$ is then fed to non-linear activation functions. When stacked together in multiple layers, such linear operations, combined with nonlinear activation functions, form the basis for more complex neural network architectures. These layers can be repeated and organized in various ways to tackle a wide range of machine learning tasks, from classification to regression, progressively transforming input data into high-level abstractions.

### 2.3 Primary objective

The goal of this work is to replace the standard linear operation in a neural network layer with a quadratic function, while keeping activation functions unchanged. Specifically, we aim to replace the linear transformation (1) with a quadratic function $g : \mathbb{R}^n \to \mathbb{R}^d$, such that the output becomes:

$$\boldsymbol{z} = g(\boldsymbol{x}; \boldsymbol{W}, \boldsymbol{\Lambda}) \tag{2}$$

where $\boldsymbol{W}$ still represents the parameters of the linear term, and $\boldsymbol{\Lambda}$ represents the parameters associated with the quadratic terms. A key challenge is to ensure that the additional matrix $\boldsymbol{\Lambda}$, responsible for quadratic terms, only adds a small number of extra parameters relative to the existing ones in $\boldsymbol{W}$, while the extra computational cost introduced by the quadratic transformation $g$ remains minimal compared to the standard linear transformation. We seek to enhance the expressiveness of the model by introducing higher-order interactions between features without significantly increasing the model's complexity or computational overhead.

## 3 Methodologies

Based on the objective outlined in Section 2.3, this section will provide a detailed description of the design of the quadratic function $g(\boldsymbol{x}; \boldsymbol{W}, \boldsymbol{\Lambda})$.

### 3.1 Quadratic transformation in a single layer

Let us begin by considering a standard quadratic transformation that introduces additional nonlinearity to the linear transformation in Equation (1) by adding a quadratic term. The resulting transformation is given by:

$$\boldsymbol{z} := \begin{bmatrix} \boldsymbol{x}^\top \boldsymbol{V}_1 \boldsymbol{x} \\ \vdots \\ \boldsymbol{x}^\top \boldsymbol{V}_d \boldsymbol{x} \end{bmatrix} + \boldsymbol{W}\boldsymbol{x} + \boldsymbol{b}, \tag{3}$$

where $\boldsymbol{V}_1, \ldots, \boldsymbol{V}_d \in \mathbb{R}^{n \times n}$ are the trainable weights for the quadratic terms, $\boldsymbol{W} \in \mathbb{R}^{d \times n}$ represents the weights for the linear term, and $\boldsymbol{b} \in \mathbb{R}^d$ is the bias vector. However, as is, this modification introduces a significant increase in the number of parameters, requiring $(dn^2)$ additional parameters, which would substantially increase the complexity of the model.

### 3.2 Rank-1 matrices for quadratic terms

A common technique to reduce the number of parameters in matrices is to impose low-rankness. Specifically, we set each matrix $\boldsymbol{V}_i, \forall i = 1, \ldots, d$, in (3) to be rank-1. That is, for two vectors

$\boldsymbol{p}_i, \boldsymbol{q}_i \in \mathbb{R}^n,$

$$V_i := \boldsymbol{p}_i \boldsymbol{q}_i^\top, \quad \forall \, i = 1, \ldots, d. \tag{4}$$

Consequently, Equation (3) becomes

$$\boldsymbol{z} = (\boldsymbol{P}\boldsymbol{x}) \odot (\boldsymbol{Q}\boldsymbol{x}) + \boldsymbol{W}\boldsymbol{x} + \boldsymbol{b}, \tag{5}$$

where $\boldsymbol{P} = [\boldsymbol{p}_1, \cdots, \boldsymbol{p}_d]^\top \in \mathbb{R}^{d \times n}$, $\boldsymbol{Q} = [\boldsymbol{q}_1, \ldots, \boldsymbol{q}_d]^\top \in \mathbb{R}^{d \times n}$ and $\odot$ denotes the Hadamard (element-wise) product. This approach effectively reduces the number of extra parameters from $(dn^2)$ to $(2dn)$, significantly lowering the model's complexity while still allowing it to capture the quadratic interactions between the input features.

### 3.3 Weight sharing

To further reduce the computational and parameter complexities of the quadratic transformation, we introduce weight sharing. Specifically, we share the weight matrix $\boldsymbol{W}$ between the linear and quadratic terms. By defining:

$$\boldsymbol{P} := \boldsymbol{\Lambda}\boldsymbol{W}, \quad \boldsymbol{Q} := \boldsymbol{W}, \tag{6}$$

where $\boldsymbol{\Lambda} \in \mathbb{R}^{d \times d}$ is a new weight matrix that differentiates the feature space of $\boldsymbol{P}$ and $\boldsymbol{Q}$, we can rewrite the quadratic transformation as:

$$\boldsymbol{z} = (\boldsymbol{\Lambda}\boldsymbol{W}\boldsymbol{x}) \odot (\boldsymbol{W}\boldsymbol{x}) + \boldsymbol{W}\boldsymbol{x} + \boldsymbol{b} = (\boldsymbol{\Lambda}\tilde{\boldsymbol{y}}) \odot \tilde{\boldsymbol{y}} + \tilde{\boldsymbol{y}} + \boldsymbol{b}, \tag{7}$$

where $\tilde{\boldsymbol{y}} := \boldsymbol{W}\boldsymbol{x}$ represents the linear transformation of the input. This reuse of the weight matrix $\boldsymbol{W}$ provides two key advantages. First, by sharing $\boldsymbol{W}$ across both the linear and quadratic components, we significantly reduce the number of model parameters. Instead of three independent parameters $\boldsymbol{W}, \boldsymbol{P}, \boldsymbol{Q} \in \mathbb{R}^{d \times n}$, the model now only needs to learn $\boldsymbol{W}$ and $\boldsymbol{\Lambda}$. Second, the linear response $\tilde{\boldsymbol{y}} = \boldsymbol{W}\boldsymbol{x}$ is computed once and reused in both the linear and quadratic operations, reducing the computational overhead.

### 3.4 Sparsification of $\boldsymbol{\Lambda}$

While the rank-1 decomposition significantly reduces the number of parameters in the quadratic term, the weight matrix $\boldsymbol{\Lambda}$ still requires $O(d^2)$ parameters, which could result in a substantial increase in model complexity. To address this, we apply a sparsification strategy to $\boldsymbol{\Lambda}$ by converting it into a band matrix. In this structure, non-zero elements are restricted to a specific "band". Additionally, we introduce two small triangular regions in the lower-left and upper-right corners, as illustrated in Figure 1. This

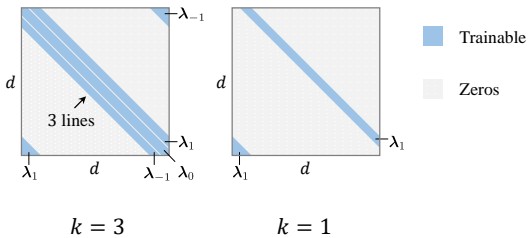

Figure 1: Sparse structure of $\boldsymbol{\Lambda}$.

allows the band matrix with width $k$ to be divided into $k$ lines of $d$-dimensional parameters, where the missing elements of these lines are filled by values from the triangular regions. Hence, the total number of trainable parameters in $\boldsymbol{\Lambda}$ is reduced to $k \times d$, with $k$ being much smaller than $d$ (e.g., $k = 1$). With this sparse structure, the computation of $\boldsymbol{\Lambda}\tilde{\boldsymbol{y}}$ becomes:

$$\boldsymbol{\Lambda}\tilde{\boldsymbol{y}} = \sum_{r \in \mathcal{K}} \boldsymbol{\lambda}_r \odot \mathrm{Roll}(\tilde{\boldsymbol{y}}, r), \tag{8}$$

where $\mathcal{K} := \{\cdots, -1, 0, 1, \cdots\}$ is the set of shifts with $|\mathcal{K}| = k$, $\boldsymbol{\lambda}_r$ represents a line of parameters in $\boldsymbol{\Lambda}$, and the function $\mathrm{Roll}(\cdot)$ is defined as

$$\mathrm{Roll}(\tilde{\boldsymbol{y}}, r) := [\tilde{y}_{1+(r \bmod d)}, \cdots, \tilde{y}_{d+(r \bmod d)}]^\top. \tag{9}$$

The advantage of using a band matrix is that it ensures the rank of $\boldsymbol{\Lambda}$ remains sufficiently high, even when the number of parameters is significantly reduced (e.g., when $k = 1$). Specifically, it guarantees that $\mathrm{rank}(\boldsymbol{P}) \leq \min\{\mathrm{rank}(\boldsymbol{\Lambda}), \mathrm{rank}(\boldsymbol{W})\}$ does not get too small, preserving the representational capacity of the quadratic term. This is crucial for maintaining the expressiveness of the model while reducing both parameter count and computational overhead. In our experiments, we exclude the shift $r = 0$, as it produces square terms $\tilde{\boldsymbol{y}}^2$, whereas any non-zero shift produces cross terms. In practice,

the square terms are more prone to numerical instabilities (such as overflows or exploding gradients) than the cross terms, especially when training with the FP16 precision whose limited dynamic range amplifies numerical instability issues. The rationale behind this design choice is illustrated in Example 3.1. Consequently, our experiments use $\mathcal{K} = \{1\}$, as shown in the right-hand subfigure of Figure 1. This choice effectively results in quadratic interactions between nearest-neighbor neurons, excluding the self-interaction.

**Example 3.1** *Let $x_1, x_2 \overset{i.i.d.}{\sim} \mathcal{N}(0,1)$ be independent standard normal random variables. The expectations and variances of the square terms $x_1^2$ and $x_2^2$, as well as the cross term $x_1 x_2$ are as follows: $E[x_1^2] = E[x_2^2] = 1, Var[x_1^2] = Var[x_2^2] = 2$ and $E[x_1 x_2] = 0, Var[x_1 x_2] = 1$. The cross term retains the expectations and variances of a normal distribution, while the square terms do not. Monte Carlo estimates in Table 1 further show that the square terms are far more likely to attain large absolute values compared to the cross terms.*

Table 1: Estimated probability of the pure quadratic and cross terms

| probability | $v = 4$ | $v = 8$ | $v = 16$ |
|---|---|---|---|
| $p(|x_1^2| > v)$ | $4.5 \times 10^{-2}$ | $4.7 \times 10^{-3}$ | $6.33 \times 10^{-5}$ |
| $p(|x_1 x_2| > v)$ | $7.6 \times 10^{-3}$ | $4.1 \times 10^{-5}$ | $3.37 \times 10^{-10}$ |

### 3.5 Quadratic enhancer

The complete workflow of the proposed quadratic enhancer is sketched in Figure 2, which is divided into two panels. We recall the conventional linear transformation with bias, $\boldsymbol{z} = \boldsymbol{W}\boldsymbol{x} + \boldsymbol{b}$, in the upper panel. The lower panel then details the quadratic enhancer itself. First, the linear transformation $\tilde{\boldsymbol{y}} = \boldsymbol{W}\boldsymbol{x}$ is computed and fed to the enhancer. A set of rolling shifts $\text{Roll}(\cdot)$ is applied to $\tilde{\boldsymbol{y}}$; these shifted copies are linearly combined by a learnable, band-sparse matrix $\boldsymbol{\Lambda}$, producing the refined response $\boldsymbol{\Lambda}\tilde{\boldsymbol{y}}$, after which a quadratically augmented feature $(\boldsymbol{\Lambda}\tilde{\boldsymbol{y}}) \odot \tilde{\boldsymbol{y}}$ is calculated. The final output is then $\boldsymbol{z} = (\boldsymbol{\Lambda}\tilde{\boldsymbol{y}}) \odot \tilde{\boldsymbol{y}} + \tilde{\boldsymbol{y}} + \boldsymbol{b}$. Figure 2 illustrates how a single linear transformation can be augmented by the quadratic enhancer. Crucially, the same enhancer block can be attached to every linear layer in a neural network, endowing the entire model with richer quadratic interactions while incurring only a negligible increase in parameters and computation.

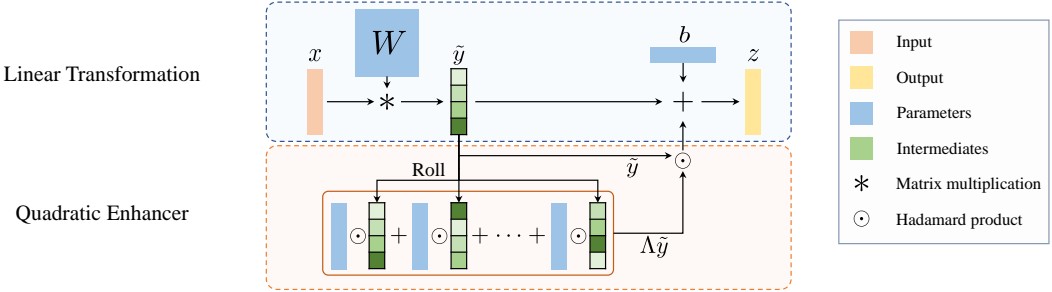

Figure 2: An overview of the quadratic enhancer.

### 3.6 Cost analysis

In this section, we evaluate the overhead introduced by the quadratic enhancement in terms of both parameters and inference FLOPs, which ultimately impact the model's efficiency.

**Parameters:** The quadratic enhancer introduces a single learnable matrix $\boldsymbol{\Lambda} \in \mathbb{R}^{d \times d}$, but with non-zero entries restricted to a bandwidth of width $k$ (where $k \ll d$, e.g., $k = 1$). This results in $k \times d$ free parameters in $\boldsymbol{\Lambda}$. In contrast, the standard linear layer employs a weight matrix $\boldsymbol{W} \in \mathbb{R}^{d \times n}$ with $d \times n$ parameters. The relative overhead in terms of parameters is thus: $\frac{kd}{nd} = O(k/n)$, which is negligible in practice given that $k \ll n$ (typically $n \approx d$).

**Inference FLOPs:** The quadratic enhancer reuses the linear activation $\tilde{\boldsymbol{y}} = \boldsymbol{W}\boldsymbol{x}$, resulting in only three additional operations. These include: (i) a matrix multiplication $\boldsymbol{\Lambda}\tilde{\boldsymbol{y}}$ which requires $2kd$ FLOPs, (ii) a Hadamard product $\odot\tilde{\boldsymbol{y}}$ which costs $d$ FLOPs, and (iii) an addition $+\tilde{\boldsymbol{y}}$ which adds another $d$ FLOPs. Therefore, the total number of extra FLOPs introduced by the quadratic enhancer is $2(k+1)d$ FLOPs. For comparison, the original linear transformation $\boldsymbol{W}\boldsymbol{x}$ requires $2nd$ FLOPs, and the bias addition adds $d$ FLOPs, leading to a total of $2nd+d$ FLOPs for the original computation. The relative overhead is thus: $\frac{2(k+1)d}{2nd+d} = O(k/n)$, which, again, becomes negligible for $k \ll n$.

## 4 Experimental results

In this section, we conduct an evaluation of the proposed quadratic enhancer across three tasks. To investigate the performance with minimal overhead, we focus on evaluating the quadratic enhancer with $\mathcal{K} = \{1\}$. All experiments were conducted using four NVIDIA A100 80GB. The experimental code is publicly available at `https://github.com/chitar/QuadEnhancer`.

### 4.1 Image classification

Image classification represents one of the most classical and foundational tasks in computer vision. It serves as a benchmark for evaluating the effectiveness of novel neural architectures and training methodologies. Following common practice, our experiments involve initial pre-training on a large-scale dataset, subsequently fine-tuning the pre-trained models on various target datasets. Specifically, we first pre-train on ImageNet-1k [20], then fine-tune and evaluate the models across several diverse downstream datasets.

**Datasets:** Our experiments begin with ImageNet-1k for the initial pre-training stage. For downstream evaluation, we use six widely recognized benchmarks: Caltech [9], CIFAR-10, CIFAR-100 [20], Flowers [32], Food [2], and Pets [33]. Caltech is a classical image recognition dataset containing various object categories. CIFAR-10 and CIFAR-100 are commonly used for small-scale visual recognition, with the former covering 10 categories and the latter 100. The Flowers dataset focuses on fine-grained classification of flower species, while the Food dataset includes a variety of food categories. The Pets dataset is centered on distinguishing different pet breeds. Further details about these datasets can be found in Table 2.

**Baselines:** The Vision Transformer (ViT) [6] serves as our baseline model due to its demonstrated effectiveness and wide adoption. By varying key hyperparameters such as hidden size and number of layers, we consider three model sizes: ViT-M, ViT-XT, and ViT-T. Specific hyperparameter configurations are detailed in the Table 3.

**Training settings:** In the pre-training phase, each baseline model and its corresponding quadratic enhancer variant are trained on ImageNet-1k. The training parameters, including

Table 2: Image datasets

| Dataset | # Classes | # Samples |
|---------|-----------|-----------|
| caltech | 102 | 9.0k |
| cifar10 | 10 | 60k |
| cifar100 | 100 | 60k |
| flower | 102 | 8.1k |
| pet | 37 | 7.4k |
| food | 101 | 101k |

batch size, learning rate, number of epochs, and total training duration, are consistent with the settings outlined in [29]. After pre-training, each model is fine-tuned individually on all downstream datasets, with performance evaluated on the corresponding validation sets.

**Results:** The experimental results are presented in Table 4, where the column labeled "ImageNet" indicates validation accuracy obtained during pre-training, and subsequent columns represent accuracy scores achieved on each downstream dataset after fine-tuning. Models incorporating the quadratic enhancer are indicated with "+QE." As shown in Table 4, models equipped with the quadratic enhancer consistently outperform their baseline counterparts across

Table 3: Model parameters of ViTs.

| | Embed_dim | Layers | FFN_dim |
|--------|-----------|--------|---------|
| ViT-M | 192 | 6 | 768 |
| ViT-XT | 128 | 12 | 512 |
| ViT-T | 192 | 12 | 768 |

all datasets. Specifically, ViT-M+QE surpasses ViT-M by $1.60\%$ on ImageNet and achieves substantial gains on downstream tasks, notably improving Caltech by $2.55\%$ and CIFAR-100 by $2.34\%$. Similarly, ViT-XT+QE and ViT-T+QE outperform their respective baselines, demonstrating significant accuracy boosts across datasets. The quadratic enhancer proves especially beneficial on the challenging Pets dataset, where ViT-XT+QE achieves an impressive $6.94\%$ improvement over

ViT-XT. Overall, these consistent performance enhancements underline the quadratic enhancer's capability to enrich model effectiveness across various visual classification tasks.

Table 4: Accuracy (%) of ViTs with and without the quadratic enhancer.

| Model | Params | ImageNet | Caltech | Cifar10 | Cifar100 | Flowers | Food | Pets | Avg |
|---|---|---|---|---|---|---|---|---|---|
| ViT-M | 2.45M | 63.70 | 87.77 | 96.35 | 80.25 | 74.01 | 83.94 | 91.03 | 82.44 |
| ViT-M+QE | 2.47M | **65.30** | **90.32** | **97.09** | **82.59** | **75.58** | **84.63** | **91.88** | **83.91** |
| ViT-XT | 2.82M | 66.04 | 90.25 | 96.51 | 81.24 | 84.41 | 85.47 | 91.03 | 84.99 |
| ViT-XT+QE | 2.83M | **67.34** | **90.77** | **96.78** | **82.64** | **86.37** | **85.85** | **97.97** | **86.82** |
| ViT-T | 5.37M | 73.96 | 93.07 | 97.97 | 86.13 | 86.56 | 88.42 | 93.87 | 88.57 |
| ViT-T+QE | 5.40M | **75.15** | **94.03** | **98.03** | **86.88** | **87.25** | **88.59** | **94.95** | **89.27** |

## 4.2 Text classification

Text classification is a cornerstone task in natural language processing, underpinning applications ranging from sentiment analysis to topic categorization. It remains a classical and fundamental benchmark for evaluating advances in language modeling and fine-tuning techniques. Analogous to image classification, modern text classification pipelines typically involve pre-training a general-purpose language model on large corpora, followed by fine-tuning on specific downstream datasets. In our experiments, we adhere to this convention: we first pre-train on a small-scale corpus, then fine-tune on several text classification benchmarks.

**Datasets:** For pre-training, we use the WikiText-2 dataset [30], a widely adopted corpus containing over 2 million tokens from English Wikipedia articles. While larger pre-training datasets exist, their computational demands exceed our resource constraints. For downstream text classification, we utilize six standard benchmarks: IMDB (movie review sentiment analysis) [27], Yelp (restaurant review sentiment) [17], AG-News (topic classification) [49], SST-2 (Stanford Sentiment Treebank) [41], and Emotion (emotion recognition) [39]. Detailed dataset statistics are available in Table 5.

**Baselines:** We use the GPT-2 architecture [36] as our baseline, given its strong performance in language modeling and its widespread applicability across various downstream tasks. We evaluate two models: GPT-2/16 and GPT-2/32, differentiated by the size of their hidden dimension.

Table 5: Text datasets.

| Dataset | # Classes | # Samples |
|---|---|---|
| IMDB | 2 | 50k |
| Yelp | 5 | 700k |
| AG-News | 4 | 120k |
| SST-2 | 2 | 67k |
| Emotion | 6 | 20k |

**Training settings:** Each model is pre-trained on WikiText-2 for 20 epochs, with batch size 128, learning rate 0.0001, and a maximum sequence length of 256 tokens, sufficient to cover most samples. Following pre-training, models are fine-tuned on each classification dataset for 10 epochs, with learning rate 0.00005, batch size 16, and other optimizer settings unchanged.

**Results:** Table 6 reports the perplexity on WikiText-2 and validation accuracy on each classification dataset. The first column lists model variants, with '+QE' indicating use of the quadratic enhancer. The second column shows the number of trainable parameters; the third column reports WikiText-2 perplexity (lower is better). Remaining columns present classification accuracy. As shown in Table 6, models augmented with the quadratic enhancer achieve consistent improvements. GPT-2/16+QE reduces perplexity from 4.90 to 4.81 and increases average accuracy by $0.91\%$, with notable gains on IMDB and Emotion datasets. GPT-2/32+QE yields a perplexity drop from 4.57 to 4.44, and boosts average accuracy by $0.86\%$, driven by significant improvements on the Emotion benchmark (from $64.50\%$ to $69.05\%$). These results demonstrate that even with minimal additional parameters, the quadratic enhancer can enhance language model expressiveness and downstream performance.

Table 6: Performance of GPT-2 variants with and without the quadratic enhancer (QE). WikiText is evaluated in perplexity (lower is better), whereas all other datasets are reported in accuracy (%).

| Model | Params | WikiText (ppl.) | IMDB | Yelp | AG-News | SST-2 | Emotion | Avg Acc. |
|---|---|---|---|---|---|---|---|---|
| GPT-2/16 | 0.71M | 4.90 | 79.68 | 93.51 | 90.90 | **79.70** | 39.70 | 76.70 |
| GPT-2/16+QE | 0.82M | **4.81** | **81.16** | **93.64** | **91.67** | 78.78 | **42.80** | **77.61** |
| GPT-2/32 | 1.56M | 4.57 | **84.01** | 93.72 | 91.97 | 81.53 | 64.50 | 83.15 |
| GPT-2/32+QE | 1.61M | **4.44** | 83.52 | **93.80** | **92.01** | **81.65** | **69.05** | **84.01** |

## 4.3 LLMs finetuning

Fine-tuning LLMs through parameter-efficient fine-tuning [11] is critically important due to its capability to efficiently adapt powerful pretrained models to diverse downstream tasks with minimal additional resources. We evaluate the quadratic enhancer's integration into the LoRA algorithm [15] and fine-tune three variants of the open-source LLaMA models [43, 44, 10].

**Dataset:** Our fine-tuning experiments focus on the commonsense reasoning task, a crucial benchmark to assess language models' practical reasoning capabilities. We use several benchmark datasets including BoolQ, PIQA, SIQA, HellaSwag, WinoGrande, ARC-e, ARC-c, and OBQA. Each dataset assesses different aspects of commonsense understanding and reasoning skills. Detailed descriptions of these datasets are provided in the appendix of [16].

**Baselines and training settings:** We select three baseline models for our experiments: LLaMA-7B [43], LLaMA2-7B [44], and LLaMA3-8B [10], reflecting a progression of model capabilities. We employ LoRA, a widely studied parameter-efficient fine-tuning method, known for its efficiency and simplicity. For each baseline, we apply the quadratic enhancer to LoRA adapters with two ranks (r=16 and r=32). Our training configurations follow the established settings from prior works [16, 26].

**Results:** The experimental results are summarized in Table 7. The first three columns specify the model names, methods (with LoRA rank indicated after the '/' and '+QE' denoting the quadratic enhancer's use), and the number of parameters trained, respectively. Subsequent columns report accuracy on the commonsense reasoning benchmark datasets. Results for LoRA without QE are taken from prior works [16, 26]. As shown in Table 7, the quadratic enhancer consistently and significantly improves performance across all model variants and benchmarks, even with half-parameter versions where the rank is 16. Notably, LLaMA2-7B with LoRA/16+QE achieves an impressive average accuracy improvement of $2.64\%$ over LoRA/32. Furthermore, LLaMA3-8B models integrated with QE outperform the baseline LoRA models, particularly on complex reasoning tasks like HellaSwag and ARC datasets. This demonstrates the quadratic enhancer's strong ability to enhance LLMs' reasoning capabilities with minimal additional computational cost and parameters.

Table 7: Accuracy (%) of LoRA finetuning on LLaMA variants with and without the quadratic enhancer (QE).

| Model | Method | Params | BoolQ | PIQA | SIQA | HellaSwag | WinoG. | ARC-e | ARC-c | OBQA | Avg |
|---|---|---|---|---|---|---|---|---|---|---|---|
| LLaMA-7B | LoRA/32 | 53.5M | 68.90 | 80.70 | 77.40 | 78.10 | 78.80 | 77.80 | 61.30 | 74.80 | 74.73 |
| | LoRA/16+QE | 27.6M | **69.69** | **82.64** | **79.68** | **87.11** | 80.11 | **79.41** | 63.99 | 80.20 | **77.85** |
| | LoRA/32+QE | 54.3M | 69.14 | 81.06 | 77.99 | 74.00 | **81.29** | 79.33 | **64.16** | 80.80 | 75.97 |
| LLaMA2-7B | LoRA/32 | 53.5M | 69.80 | 79.90 | 79.50 | 83.60 | 82.60 | 79.80 | 64.70 | 81.00 | 77.61 |
| | LoRA/16+QE | 27.6M | **72.26** | **82.86** | 79.78 | 86.98 | **83.66** | **85.35** | 68.51 | **82.60** | **80.25** |
| | LoRA/32+QE | 54.3M | 69.63 | 82.53 | **80.24** | **90.01** | 83.03 | 83.41 | 68.51 | 80.80 | 79.77 |
| LLaMA3-8B | LoRA/32 | 54.0M | 70.80 | 85.20 | 79.90 | 91.70 | 84.30 | 84.20 | 71.20 | 79.00 | 80.79 |
| | LoRA/16+QE | 27.7M | 74.40 | 88.46 | 80.29 | **95.45** | 86.42 | **91.37** | **80.63** | 85.00 | 85.25 |
| | LoRA/32+QE | 54.7M | **74.92** | **89.44** | **81.32** | 95.02 | **87.29** | 89.85 | 79.60 | **86.20** | **85.46** |

## 4.4 Additional experimental results

In addition to the main results of the three tasks considered, we perform several further experiments to evaluate the performance and scalability of the proposed quadratic enhancer. These experiments provide further insights into its comparative advantages and practical considerations in different settings. The results of these additional experiments are reported below.

**Comparison with quadratic baselines**   We additionally compare QuadEnhancer with two existing quadratic MLP variants: (i) **QuadraNet** [48], which uses the quadratic formulation $y = W_a x \odot W_b x + W_c x$, where $W_a, W_b, W_c \in \mathbb{R}^{d \times n}$ are learnable parameters, and (ii) **SwiGLU** [40], which defines the quadratic interaction as $y = (W_1 x) \odot \text{Sigmoid}(W_1 x) \odot (W_2 x)$, with $W_1, W_2 \in \mathbb{R}^{d \times n}$. All experiments are conducted on the image classification task using the ViT-M model as the backbone, and the number of parameters is matched by adjusting the hidden dimension $d$. The results, presented in Table 8, demonstrate that QuadEnhancer consistently outperforms both quadratic baselines across all evaluated datasets.

Table 8: Accuracy (%) of different quadratic methods.

| Method | Param | Imagenet1k | Cifar10 | Cifar100 | Food | Avg |
|---|---|---|---|---|---|---|
| ViT-M+QuadraNet | 2.53M | 61.17 | 95.81 | 79.08 | 81.58 | 79.41 |
| ViT-M+SwiGLU | 2.58M | 63.25 | 96.76 | 80.58 | 83.91 | 81.13 |
| ViT-M+QuadEnhancer | 2.47M | **65.30** | **97.09** | **82.59** | **84.63** | **82.40** |

**Scaling behavior**   Understanding the scaling behavior of a model is essential for evaluating its effectiveness as both data and model sizes increase. While large-scale experiments were not feasible due to computational constraints, we conduct experiments on data and model sizes ranging from small to moderate scales. These experiments offer insights into the scaling behavior of the proposed quadratic enhancer. All experiments are conducted on the image classification task with varying hidden dimensions $d$ and dataset sizes $s$. As shown in Table 9, we observe that the quadratic enhancer yields increasing performance gains as both the model and dataset sizes grow.

Table 9: Accuracy (%) of different scales.

| | $d = 24, s = 50k$ | $d = 48, s = 100k$ | $d = 96, s = 200k$ | $d = 192, s = 400k$ |
|---|---|---|---|---|
| ViT | 8.99 | 19.18 | 33.14 | 49.59 |
| ViT+QE | 9.06 | 19.95 | 34.03 | 50.78 |
| Gain | **0.07** | **0.77** | **0.89** | **1.19** |

**Training time comparison**   To evaluate the trade-offs between computational cost and performance, we report the training times for two tasks: pretraining on ImageNet-1k (Table 10) and fine-tuning on CIFAR-100 (Table 11). The results show that while the quadratic-enhanced models take slightly longer to converge in the early stages of training, they ultimately achieve superior performance. These findings suggest that while QuadEnhancer introduces some initial overhead, it offers substantial long-term performance gains.

Table 10: Accuracy (%) of pretraining on ImageNet-1k at different time.

| Method | Time(sec.) | | | | | |
|---|---|---|---|---|---|---|
| | 1k | 10k | 20k | 30k | 40k | 50k |
| ViT | **22.28** | **52.09** | 58.35 | **63.51** | 65.96 | 65.97 |
| ViT+QE | 11.30 | 51.28 | **59.07** | 63.15 | **66.59** | **67.04** |

Table 11: Accuracy (%) of finetuning on CIFAR-100 at different time.

| Method | Time(sec.) | | | | | | |
|---|---|---|---|---|---|---|---|
| | 500 | 1k | 2k | 3k | 4k | 5k | 6k |
| ViT | 78.79 | **81.78** | 83.00 | 84.21 | 85.29 | 85.29 | 85.29 |
| ViT+QE | **79.33** | 81.07 | **83.43** | **84.49** | **85.48** | **86.15** | **86.53** |

**Ablation study on $\mathcal{K}$**   We conduct an ablation study to investigate the effect of different choices for the set $\mathcal{K}$ in image classification tasks. Models with various $\mathcal{K}$ sets and hidden dimensions are trained

from scratch on the Caltech dataset using ViT-M as the backbone. The accuracy results, shown in Table 12, indicate that increasing the size of $\mathcal{K}$ generally improves performance. The most significant improvement occurs when moving from $\mathcal{K} = \emptyset$ to $\mathcal{K} = \{1\}$, with further increases in $\mathcal{K}$ yielding diminishing returns. This suggests a clear trade-off between model complexity and performance.

Table 12: Accuracy (%) when using different $\mathcal{K}$ and hidden size.

| $\mathcal{K}$ | Hidden dimension | | | | | |
|---|---|---|---|---|---|---|
| | 24 | 48 | 96 | 144 | 192 | 288 |
| $\emptyset$ | 47.16 | 54.02 | 58.20 | 59.87 | 60.50 | 61.13 |
| $\{1\}$ | 48.94 | 54.79 | 59.35 | 60.83 | 61.39 | 61.57 |
| $\{-1, 1\}$ | 49.38 | 55.09 | 59.83 | 61.28 | 61.80 | 61.65 |
| $\{-2, -1, 1, 2\}$ | 49.72 | 55.16 | 60.28 | 61.39 | 61.50 | 61.76 |

## 5  Limitation and conclusions

**Limitation:** A key limitation of the quadratic enhancer is that it introduces additional computational overhead, which, while minimal in terms of parameters and FLOPs, still represents an extra cost compared to traditional linear transformations. However, our approach effectively minimizes this impact, ensuring that the performance improvements far outweigh the added complexity.

**Conclusions:** In this work, we construct a class of quadratic enhancers to enable quadratic interactions among features at a neural network layer. Compared to a fully-connected linear transformation, the quadratic enhancers require only negligible amounts of extra parameters and FLOPs. Our proof-of-concept experiments, conducted across multiple tasks and multiple datasets, have confirmed the potentials of the proposed approach in delivering substantial performance improvements without notably increasing model sizes. Evidences even suggest that, in some cases, our approach may be able to significantly reduce model sizes while maintaining or even enhancing performance levels. At this point, however, our study is still quite preliminary. More research is definitely needed in this direction to better understand the promises and limitations of the proposed approach.

## Acknowledgments

This work was supported by the National Key R&D Program of China under grant 2023YFA1009300, and by the Shenzhen Science and Technology Program (Grant No. GXWD20201231105722002-20200901175001001). Qian Chen, Linxin Yang, and Akang Wang also acknowledge support from the National Natural Science Foundation of China (Grant No. 12301416) and the Guangdong Basic and Applied Basic Research Foundation (Grant No. 2024A1515010306). Xiaodong Luo also acknowledges support from the Hetao Shenzhen-Hong Kong Science and Technology Innovation Cooperation Zone Project (No. HZQSWS-KCCYB-2024016).

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
