# OpenReview forum: "QuadEnhancer: Leveraging Quadratic Transformations to Enhance Deep Neural Networks"
_NeurIPS.cc/2025/Conference — NeurIPS 2025 poster_

### Official Review · Reviewer_fG69 · 2025-06-25

**Clarity:** 3
**Significance:** 3
**Originality:** 2
**Rating:** 4
**Confidence:** 3

**Summary:**

This paper introduces "QuadEnhancer," a lightweight module designed to enhance the representational power of deep neural networks by introducing quadratic feature interactions. The core idea is to augment standard linear transformations with a quadratic term. To make this computationally feasible and parameter-efficient, the authors propose a series of design choices: (1) reuse the output of the linear transformation to form the quadratic term, avoiding redundant computations; (2) parameterize the quadratic interactions via a new matrix; and (3) make the matrix extremely sparse, modeling it as a band matrix that can be implemented with a few efficient circular shift operations. In final implementation, this adds only d parameters (where d is the feature dimension) and negligible FLOPs per layer. The authors demonstrate the effectiveness of QuadEnhancer through experiments on image classification (ViT), text classification (GPT-2), and parameter-efficient fine-tuning of large language models (LLaMA and LoRA), showing performance gains across tasks and model scales.

**Questions:**

Please see weaknesses described above.

**Ethical Concerns:**

["NO or VERY MINOR ethics concerns only"]

**Final Justification:**

My final justification stays the same as above original review.

**Limitations:**

yes

**Paper Formatting Concerns:**

no concern

**Quality:**

3

**Strengths And Weaknesses:**

Strengths:

The proposed QuadEnhancer is a simple, general, and seemingly effective method. It acts as a "drop-in" replacement for standard linear layers and is applicable to a wide range of architectures (demonstrated on Transformers) and modalities (vision and language). The consistent performance improvements with minimal overhead suggest it could become a standard component in future model design, much like novel activation functions or normalization layers.

The paper's claims are backed by a convincing set of experiments. The authors evaluate their method on three distinct, important tasks (image classification, autoregressive language modeling, and LLM fine-tuning). The choice of strong, modern baselines (ViT, GPT-2, LLaMA) and standard datasets makes the results credible.

The paper is well-written and easy to follow.

Weaknesses:

The choice of K={1} is well-motivated for efficiency, but its impact relative to other choices (e.g., K={-1, 1}) is not explored. An ablation on K would provide a clearer picture of the trade-off between model complexity and performance.

The decision to exclude the squared term (r=0) is justified by potential FP16 instability. However, a direct empirical comparison (e.g., training with and without this term in both FP16 and FP32) is needed to validate this claim and quantify its effect on both stability and final accuracy. In addition, recently BF16 becomes more popular and is believed to have better stability than FP16.

The paper is heavily empirical. While the motivation is sound and the results are good, the paper would benefit from a deeper discussion on why these specific local quadratic interactions are effective. For example, a theoretical analysis of how this enhancement affects the function class that the network can approximate would provide more fundamental understanding and elevate the work.

---

> ### Author Rebuttal · Authors · 2025-07-31
>
> Thank you for your kind support and valuable suggestions. Some of the additional experiments are still in progress and could not be completed within the rebuttal period. We will update the results and provide clarification in the discussion stage. We apologize for any inconvenience this may cause.

---

> > ### Author Response · Authors · 2025-08-01
> > **update**
> >
> > We apologize for the delayed update. Please find the detailed clarifications below.
> >
> > > W1: The choice of K={1} is well-motivated for efficiency, but its impact relative to other choices (e.g., K={-1, 1}) is not explored. An ablation on K would provide a clearer picture of the trade-off between model complexity and performance.
> >
> > Thank you for your valuable suggestion. We have added an ablation study on different choices of K for image classification. Models with varying K sets and hidden dimensions were trained from scratch on the Caltech dataset. Accuracy ($\\uparrow$) results are shown below:
> >
> > |                | 24    | 48    | 96    | 144   | 192   | 288   |
> > |----------------|-------|-------|-------|-------|-------|-------|
> > | $K=\\emptyset$      | 47.16 | 54.02 | 58.20  | 59.87 | 60.50  | 61.13 |
> > | $K=\\{1\\}$        | 48.94 | 54.79 | 59.35 | 60.83 | 61.39 | 61.57 |
> > | $K=\\{-1,1\\}$     | 49.38 | 55.09 | 59.83 | 61.28 | 61.80  | 61.65 |
> > | $K=\\{-2,-1,1,2\\}$ | 49.72 | 55.16 | 60.28 | 61.39 | 61.50  | 61.76 |
> >
> > From the results, we observe:
> > 1. Increasing the size of K generally leads to better performance.
> > 2. The largest gain occurs from $K=\\emptyset$ to $K=\\{1\\}$, while further improvements diminish as $|K|$ grows—indicating a clear trade-off between model complexity and performance.
> >
> > > W2: The decision to exclude the squared term (r=0) is justified by potential FP16 instability. However, a direct empirical comparison (e.g., training with and without this term in both FP16 and FP32) is needed to validate this claim and quantify its effect on both stability and final accuracy. In addition, recently BF16 becomes more popular and is believed to have better stability than FP16.
> >
> >
> > Thank you for your thoughtful comment. In our early experiments using K={0}, we occasionally observed training instability—specifically, NaN losses—emerging during the mid-to-late stages of training. Switching to full FP32 (from mixed FP16) reduced but did not eliminate this issue. In contrast, using only cross terms consistently led to stable training across all runs.
> >
> > However, the instability is not fully reproducible and varies across runs and tasks, making it difficult to provide a controlled comparison within the limited rebuttal period. We appreciate your insightful points regarding FP16, FP32, and BF16, and will keep them in mind for future investigations.
> >
> >
> > > W3: The paper is heavily empirical. While the motivation is sound and the results are good, the paper would benefit from a deeper discussion on why these specific local quadratic interactions are effective. For example, a theoretical analysis of how this enhancement affects the function class that the network can approximate would provide more fundamental understanding and elevate the work.
> >
> >
> > Thank you for this insightful suggestion. We agree that a theoretical analysis of local quadratic interactions would deepen understanding, and we plan to pursue this in future work.

---

### Official Review · Reviewer_yMZu · 2025-06-27

**Clarity:** 4
**Significance:** 4
**Originality:** 4
**Rating:** 4
**Confidence:** 4

**Summary:**

The paper proposes QuadEnhancer, a lightweight module designed to enhance non-linearity through quadratic transformation while maintaining low computational complexity. This is achieved via a modular design that can be applied to any layer in a neural network. To reduce the complexity typically associated with quadratic transformations, the authors introduce three key techniques: (1) Low-rank matrices for the quadratic terms, (2) Weight sharing, and (3) Sparsification of the weight matrix. For low-rankness, each weight matrix in the quadratic term is decomposed into the outer product of two vectors, p and q, effectively constraining it to rank 1. To further reduce the number of parameters, the weight matrix W used in the linear term is also reused in the quadratic term's P and Q matrices. Lastly, the matrix Λ, which complements W to form P, is sparsified as a band matrix, maintaining expressiveness while further reducing parameter count. By integrating QuadEnhancer into various models, the authors demonstrate consistent and efficient performance improvements across a range of tasks.

**Questions:**

Please refer to the Weaknesses part.

**Ethical Concerns:**

["NO or VERY MINOR ethics concerns only"]

**Final Justification:**

The proposed method is clearly described, and the modular design is well structured. However, my concerns were not fully addressed during the rebuttal period. After reading the other reviewers' comments and the authors' response, I noticed a few points that remain unresolved.

There is no empirical comparison with gated MLP variants like GEGLU or SwiGLU, which are widely used and relevant. While I understand that a fully equivalent comparison may not be possible, including them as baselines or providing ablation results would make the evaluation more robust.

Additionally, the comparison with prior quadratic models seems somewhat overgeneralized, possibly due to the lack of a related work section discussing how other approaches have addressed efficiency. Including such context would strengthen the paper.

For these reasons, I adjust my score to 4.

**Limitations:**

Yes

**Quality:**

4

**Strengths And Weaknesses:**

**Strengths**
- The paper is clearly written and easy to follow.
- The proposed method is well-formulated with a modular structure, allowing it to be easily integrated into various architectures.
- The method is evaluated across diverse tasks and backbone models, demonstrating its generality and practical utility.

**Weaknesses**
- While the paper presents several tricks to reduce complexity, it would be more convincing to include a complexity–accuracy trade-off graph, under settings with smaller models or limited data.
- The experiment could be strengthened by providing a more explicit comparison with prior work in terms of both performance and computational cost.

---

> ### Author Rebuttal · Authors · 2025-07-31
>
> Thank you for your kind support and valuable suggestions. Some of the additional experiments are still in progress and could not be completed within the rebuttal period. We will update the results and provide clarification in the discussion stage. We apologize for any inconvenience this may cause.

---

> > ### Author Response · Authors · 2025-08-01
> > **update**
> >
> > We apologize for the delayed update. Please find the detailed clarifications below.
> >
> > > #### W1: While the paper presents several tricks to reduce complexity, it would be more convincing to include a complexity–accuracy trade-off graph, under settings with smaller models or limited data.
> >
> >
> > Thank you for your valuable suggestions. We have added an experiment for image classification. Models with varing K sets and hidden dimensions are trained on the caltech dataset from scratch. The results are shown below:
> >
> > |                | 24    | 48    | 96    | 144   | 192   | 288   |
> > |----------------|-------|-------|-------|-------|-------|-------|
> > | $K=\\emptyset$      | 47.16 | 54.02 | 58.20  | 59.87 | 60.50  | 61.13 |
> > | $K=\\{1\\}$        | 48.94 | 54.79 | 59.35 | 60.83 | 61.39 | 61.57 |
> > | $K=\\{-1,1\\}$     | 49.38 | 55.09 | 59.83 | 61.28 | 61.80  | 61.65 |
> > | $K=\\{-2,-1,1,2\\}$ | 49.72 | 55.16 | 60.28 | 61.39 | 61.50  | 61.76 |
> >
> > From the results, we observe:
> > 1. Increasing the size of K generally leads to better performance.
> > 2. The largest gain occurs from $K=\emptyset$ to $K=\\{1\\}$, while further improvements diminish as $|K|$ grows—indicating a clear trade-off between model complexity and performance.
> >
> > > #### W2: The experiment could be strengthened by providing a more explicit comparison with prior work in terms of both performance and computational cost.
> >
> > Thank you for your thoughtful suggestion. We additionally reported the results of accuracy and training time (reflecting the computational cost) on a pretraining and a finetuning task for image classification.
> >
> > Pretraining on ImageNet-1k
> > |time(sec.)|1k|10k|20k|30k|40k|50k|
> > |-|-|-|-|-|-|-|
> > |ViT|**22.28**|**52.09**|58.35|**63.51**|65.96|65.97|
> > |ViT+QE|11.30|51.28|**59.07**|63.15|**66.59**|**67.04**|
> >
> > Finetuning on Cifar100
> > |time(sec.)|500|1k|2k|3k|4k|5k|6k|
> > |-|-|-|-|-|-|-|-|
> > |ViT|78.79|**81.78**|83.0|84.21|85.29|85.29|85.29|
> > |ViT+QE|**79.33**|81.07|**83.43**|**84.49**|**85.48**|**86.15**|**86.53**|
> >
> > These results show that ViT+QE achieves better accuracy with lower computational cost in the later training stages.

---

> > ### Comment · Reviewer_yMZu · 2025-08-05
> >
> > Since the rebuttal was submitted after the official NeurIPS deadline, it will not be considered in the scoring process, in accordance with the NeurIPS official rules. The final score will be determined based on the other reviewers’ comments and the responses provided within the allowed rebuttal period.

---

> > > ### Author Response · Authors · 2025-08-08
> > > **Thank you**
> > >
> > > Thank you very much for your support and for taking the time to engage with our work. We fully understand and respect the NeurIPS guidelines regarding the rebuttal timeline. We truly appreciate thoughtful comments during the review process.

---

### Official Review · Reviewer_rPUm · 2025-06-30

**Clarity:** 4
**Significance:** 3
**Originality:** 3
**Rating:** 4
**Confidence:** 2

**Summary:**

Adds a single banded matrix of parametric terms to the traditional linear model of parameters for a neural network (that is kd in parameters where d is the number of inputs and k is small (16/32 for LoRA)). With this small increase in parameters it enables quadratic terms to be included on top of standard neural network linear models. Shows a series of experiments on different tasks and datasets showing that this is effective.

**Questions:**

1)	Empirical comparison of recent non-linear neural network approaches rather than just a linear baseline.
2)	Can QuadraNet V2 be simply dismissed by computational number of parameters? Is this accurate more broadly for the other papers stated?
3)	Empirical FLOPS comparison.
4)	Does the result hold up over broader training conditions? The experimental results have not clearly established this. Conditions shown are too narrow to be confident that the results are generally strong. Especially, does the result generalize for epochs, are they stable?

**Ethical Concerns:**

["NO or VERY MINOR ethics concerns only"]

**Final Justification:**

Thanks to the authors for a thorough rebuttal. It answers all the points that I raised.

I support this paper.

**Paper Formatting Concerns:**

N/A.

**Quality:**

3

**Strengths And Weaknesses:**

Strengths:
1)	Clear presentation of approach in a directly implementable format of vector multiplication.
2)	Well-written, well-presented, clear and appears novel (see caveat in W2).
3)	Approach is forced towards cross terms, not squared terms for numeric stability.
4)	Using VIT as baseline, on some smaller datasets and re-training LLMs, it shows solid gains given small increases in parameters sizes. This is consistent across a range of problems.

Weaknesses:
1)	Although it is shown in the introduction that there are other attempts of quadratic based neural network models, including recently, the paper does not offer any experimental comparisons to these. Different architectures may present some challenges here, but this is not discussed. Comparison only include baselines (only VIT and GPT-2 variants), no competitor networks that have quadratic or any additional parametric approaches.
2)	The claim in the introduction – on line 60 that QuadraNet V2 has substantial increases computation and parameters does not seem accurate. The QuadraNet V2 paper uses constant training time and shows small increases in number of parameters vs baselines in its experiments. (Maybe it has compute time increase, but the experiments offset this). Please comment. I may have mis-understood this. It raises a slightly broader question of whether this critique is correct of the other previous literature also.
3)	There is no comparison in terms of fixed parameter numbers, or fixed training times. (Although the tables show large increases proportional to parameter increases).
4)	The FLOPS comparisons are all theoretical (which are very nice numbers in these comparisons), however, the experimental FLOPS are not stated so we don't get to see if the theoretical model can be implemented in this efficient fashion.
5)	Training details only follow a single model from another paper for image classification, and just 20 epochs pre-training, then 10 epochs fine tuning on text classification datasets. This seems a little narrow to understand whether these are sustained results with different numbers of epochs. Are the results robust?
6)	It is slightly surprising that in Tab 8, the 16 rank outperforms 32 in the majority of cases for LLaMA{_,2}-7B. What is the cause of this?

Overall the paper appears good, however, its not clear that the authors have sufficiently established its empirical strength over other approaches to increasing the parameterisation of the standard linear activation model.

---

> ### Author Rebuttal · Authors · 2025-07-31
>
> > #### W1&Q1: Although it is shown in the introduction that there are other attempts of quadratic based neural network models, including recently, the paper does not offer any experimental comparisons to these. Different architectures may present some challenges here, but this is not discussed. Comparison only include baselines (only ViT and GPT-2 variants), no competitor networks that have quadratic or any additional parametric approaches.
>
> Thank you for your valuale comment. We have added comparisons on image classification with QuadraNets, where the quadratic term is expressed as $y = (x @ W_1)*(x@W_2) + x@W_3$. To ensure fairness, we carefully matched the number of trainable parameters between QuadraNet (QN) and our QuadEnhancer(QE) by adjusting the size of hidden dimensions. The accuracy across several standard benchmarks is summarized below and show that QE significantly outperforms QN in general:
>
>
> |            | params(M) | imagenet1k | caltech | cifar10 | cifar100 | flowers |  food  |   pet  |   avg  |
> |:----------:|:---------:|:----------:|:-------:|:-------:|:--------:|:-------:|:------:|:------:|:------:|
> | ViT-M + QN |   5.47    |   72.15    |  92.14  |  97.79  |  85.36   |  75.39  | 86.83  | 94.05  | 86.24  |
> | ViT-M + QE |   5.40    |   75.15    |  94.03  |  98.03  |  86.88   |  87.25  | 88.59  | 94.95  | 89.27  |
> | ViT-T + QN |   2.53    |   61.17    |  85.36  |  95.81  |  79.08   |  34.50  | 81.58  | 98.31  | 76.54  |
> | ViT-T + QE |   2.47    |   65.30    |  90.32  |  97.09  |  82.59   |  75.58  | 84.63  | 91.88  | 83.91  |
>
> Thank you for your comment again, we have added these comparisons to the revised manuscript.
>
> > #### W2&Q2: The claim in the introduction – on line 60 that QuadraNet V2 has substantial increases computation and parameters does not seem accurate. The QuadraNet V2 paper uses constant training time and shows small increases in number of parameters vs baselines in its experiments. (Maybe it has compute time increase, but the experiments offset this). Please comment. I may have mis-understood this. It raises a slightly broader question of whether this critique is correct of the other previous literature also.
>
> Thank you for your thoughtful observation, and we apologize for the confusion caused by our earlier wording. Our intent was not to suggest that QuadraNets and other work specifically results in prohibitively large training time or parameter counts than linear NNs, but rather to highlight that the parameter and computational complexity of the quadratic components in prior quadratic models is generally higher than that of QuadEnhancer. For example, models in [3,4] typically involve quadratic terms with $O(n^2)$ parameters and computation (some even up to $O(2n^2)$ [1,2,5]), while QuadEnhancer introduces only $O(n)$ parameters and computational overhead for the quadratic term (with $k=1$).
>
> We appreciate you raising this point and have added this clarification in the revised the manuscrip.
>
> > #### W3: There is no comparison in terms of fixed parameter numbers, or fixed training times. (Although the tables show large increases proportional to parameter increases).
>
> Thank you for your insightful comment. We understand the importance of comparisons with fixed parameter counts or fixed training times. To address this:
> 1. Parameter Comparison: The additional number of parameters introduced by the quadratic models is relatively small compared to their linear counterparts. For example, in the text classification task, we observe the parameter counts are 2.47M vs. 2.45M, 2.83M vs. 2.82M, and 5.40M vs. 5.37M. Given the small difference in parameters, we did not perform a comparison based strictly on identical parameter numbers, as the changes are minimal.
> 2. Training Time Comparison: To provide further insight into the trade-offs, we have reported training times for two tasks: pretraining on ImageNet-1k and fine-tuning on CIFAR100. The results show that the models with QuadEnhancer (ViT+QE) generally outperform their linear counterparts (ViT) in the later stages of training. Below are the results:
>
> Pretrainign on ImageNet-1k
> |time(sec.)|1k|10k|20k|30k|40k|50k|
> |-|-|-|-|-|-|-|
> |ViT|**22.28**|**52.09**|58.35|**63.51**|65.96|65.97|
> |ViT+QE|11.30|51.28|**59.07**|63.15|**66.59**|**67.04**|
>
> Finetuning on Cifar100
> |time(sec.)|500|1k|2k|3k|4k|5k|6k|
> |-|-|-|-|-|-|-|-|
> |ViT|78.79|**81.78**|83.0|84.21|85.29|85.29|85.29|
> |ViT+QE|**79.33**|81.07|**83.43**|**84.49**|**85.48**|**86.15**|**86.53**|
>
>
> > #### W4&Q3: The FLOPS comparisons are all theoretical (which are very nice numbers in these comparisons), however, the experimental FLOPS are not stated so we don't get to see if the theoretical model can be implemented in this efficient fashion.
>
>
> Thank you for your comment.  We measured the actual number of FLOPs using NVIDIA's Nsight Compute. We recorded the executed FLOPs for both a standard linear layer and a quadratic-enhanced (QE) layer, each with hidden dimension n=192, averaged over 64 runs. The results are:
>
> | Linear |   QE  |
> |--------|:-----:|
> | 93888  | 94848 |
>
> This indicates that QE introduces only a ~1.0% increase in actual FLOPs. The theoretical increase is approximately 1/192=0.5%, so the gap is small and reasonable. We note that our current implementation does not yet leverage low-level CUDA or Triton optimization, which could further close this gap.
>
>
>
> > #### W5&Q4:  Training details only follow a single model from another paper for image classification, and just 20 epochs pre-training, then 10 epochs fine tuning on text classification datasets. This seems a little narrow to understand whether these are sustained results with different numbers of epochs. Are the results robust?
>
> Thank you for your comment. We clarify below:
>
> 1. For image classification, the hyper-parameters followed [7]. The authors of [7] have made efforts to tune the hyper-parameters, thus we didn't change their settings. The training runs for hundreds of epochs (300), and the results are stable in the later stages, indicating robustness.
>
> 2. For text classification, we used 20 epochs because the model's perplexity had already converged by then. As shown below, further training beyond 16 epochs yields minimal improvement:
>
> |             | 1        |   4   |   8   |   12  |   16  |   20  |
> |-------------|----------|:-----:|:-----:|:-----:|:-----:|:-----:|
> | GPT-2/16    | 34.47    | 5.58  | 5.16  | 5.00  | 4.90  | 4.90  |
> | GPT-2/16+QE | 6.42     | 5.31  | 5.00  | 4.90  | 4.81  | 4.81  |
> | GPT-2/32    | 6.42     | 5.10  | 4.76  | 4.62  | 4.57  | 4.57  |
> | GPT-2/32+QE | 5.64     | 4.85  | 4.62  | 4.48  | 4.44  | 4.44  |
>
> And the finetuning of text classifiction is relatively simpler than the pretraining, thus we used 10 epoches, also yielding a similar coverged results at late training stage.
>
>
> > #### W6: It is slightly surprising that in Tab 8, the 16 rank outperforms 32 in the majority of cases for LLaMA{_,2}-7B. What is the cause of this?
>
> Thank you for your question. The performance drop is due to overfitting. Acctually, this phenomenon also occured in the baseline [6] when the rank was increased to 64. QuadEnhancer, by enhancing the model's capacity, causes overfitting to occur earlier, rather than worsening it.
>
>
> ----
> [1] Xu, Chenhui, et al. "Quadranet: Improving high-order neural interaction efficiency with hardware-aware quadratic neural networks." 2024 29th Asia and South Pacific Design Automation Conference (ASP-DAC). IEEE, 2024.
>
> [2] Xu, Chenhui, et al. "Quadranet v2: Efficient and sustainable training of high-order neural networks with quadratic adaptation." arXiv preprint arXiv:2405.03192 (2024).
>
> [3] Chrysos, Grigorios G., et al. "Deep polynomial neural networks." IEEE transactions on pattern analysis and machine intelligence 44.8 (2021): 4021-4034.
>
> [4] Fan, Feng-Lei, et al. "On expressivity and trainability of quadratic networks." IEEE Transactions on Neural Networks and Learning Systems (2023).
>
> [5] Fan, Feng-Lei, et al. "One neuron saved is one neuron earned: On parametric efficiency of quadratic networks." IEEE Transactions on Pattern Analysis and Machine Intelligence (2025).
>
> [6] Hu, Zhiqiang, et al. "Llm-adapters: An adapter family for parameter-efficient fine-tuning of large language models." arXiv preprint arXiv:2304.01933 (2023).
>
> [7] Mehta, Sachin, Farzad Abdolhosseini, and Mohammad Rastegari. "Cvnets: High performance library for computer vision." Proceedings of the 30th ACM International Conference on Multimedia. 2022.

---

> > ### Comment · Reviewer_rPUm · 2025-08-04
> > **Thanks for the rebuttal**
> >
> > Thanks to the authors for a thorough rebuttal. It answers all the points that I raised.
> >
> > I support this paper and will increase my score.

---

> > > ### Author Response · Authors · 2025-08-08
> > > **Thank you for your support**
> > >
> > > We sincerely thank you for your thoughtful feedback and your support of our work. Your comments were valuable in helping us strengthen the paper.

---

### Official Review · Reviewer_gte1 · 2025-07-02

**Clarity:** 3
**Significance:** 2
**Originality:** 2
**Rating:** 3
**Confidence:** 4

**Summary:**

The paper addresses the problem of improving the expressiveness of neural network architectures by introducing a new feed-forward dense layer design called QuadEnhancer. This module augments traditional dense layers with structured quadratic interactions between features, implemented efficiently via low-rank decompositions and weight sharing. The proposed layer is designed to be a lightweight drop-in replacement for standard MLP layers, providing enhanced modeling capacity with minimal additional cost. The authors present an efficient implementation of QuadEnhancer and analyze some of its theoretical properties. Empirical evaluations include experiments on image classification benchmarks as well as language model fine-tuning, demonstrating improved performance across both domains.

**Questions:**

1. Am I correct that the ViT architecture used in your experiments has a two-layer MLP with GELU as the nonlinearity, and does not use SwiGLU or GEGLU? If so, why did you choose this configuration, given that GELU-based MLPs are used in older models, and most modern architectures—including recent LLMs—use SwiGLU instead?

2. Do you have results on models of different sizes? I understand that scaling up may be costly, but even a sweep over smaller models would help clarify how the benefit of your method depends on model size. A plot showing the gain from QuadEnhancer as a function of model size (under a standardized setup) would be very helpful and reduce concerns about cherry-picked results.

3. Could you clarify how exactly you combined your method with LoRA? If the base architecture remains unchanged (e.g., still uses a two-layer GELU MLP), how does the quadratic enhancement interact with the low-rank LoRA update? Did you modify the `A @ B` structure to insert a quadratic term between them?

**Ethical Concerns:**

["NO or VERY MINOR ethics concerns only"]

**Final Justification:**

I remain unconvinced that this work is meaningfully different from many similar papers proposing alternative nonlinearities. Given the popularity of the question “what other nonlinearity can we plug in to improve networks,” I find it inadequate to present a single such nonlinearity, show benchmark results, and base a paper on that. In general, I would score such work as 2, but since the paper is well written and the benchmarks are adequate, I will keep my original score of 3: Borderline reject.

**Limitations:**

I agree with authors

**Paper Formatting Concerns:**

no formatting conceerns

**Quality:**

3

**Strengths And Weaknesses:**

**Strengths**

1. The proposed layer introduces virtually no overhead. The number of additional parameters is negligible, and the computational cost should be minimal in practice. Despite being noted as a limitation, the overhead is likely irrelevant: pointwise nonlinearities after standard dense layers are memory-bound, and with a small $k$, a well-optimized CUDA or Triton kernel fusing the quadratic interaction with the linear and bias terms should match the speed of a standard dense layer.
2. Experimental results show consistent improvements across multiple architectures and domains, including both image classification and LLM fine-tuning tasks.
3. The code for all experiments is provided, which supports reproducibility. I have not personally run or verified the code, but the effort to share it is appreciated.

 **Weaknesses**

1. No mention of gated linear units: The paper does not mention or compare with widely-used gated MLP variants like [GEGLU](https://arxiv.org/abs/2002.05202) and [SwiGLU](https://arxiv.org/abs/2002.05202), which also introduce quadratic interactions. For example, SwiGLU can be expressed as:

    - `w1_out = X @ W1`
    - `w2_out = X @ W2`
    - `SwiGLU = w1_out * sigmoid(w1_out) * w2_out`
        This includes multiplicative terms between transformed inputs and is used in most modern LLMs. Ignoring such strong and relevant baselines is a serious omission.
2. Hard to assess how meaningful the improvement is: The field is full of small changes to MLP layers, and it's usually possible to show improvement by carefully choosing tasks, hyperparameters, or model sizes. The paper does not provide state-of-the-art results on any benchmark. A method that truly adds value would likely stand out more clearly across many tasks and domains.
3. No analysis of scaling behavior: Architectural changes often show benefits on small models but lose their impact as model size increases. The experiments in the paper use relatively small-scale models and do not demonstrate behavior on wider range of scales.

---

> ### Author Rebuttal · Authors · 2025-07-31
>
> > #### W1: No mention of gated linear units
>
> Thank you for your insightful comment. We have added clarifications in the revised version, and respond to your concerns in more detail below:
> 1. Complementary, not competitive.
>     - Gated MLP variants like GEGLU and SwiGLU are not direct competitors to QuadEnhancer, and thus are not appropriate baselines for comparison. These gated variants are typically applied within FFNs/MLPs, and involve combinations with activation functions. In contrast, **QuadEnhancer is a general mechanism that can be applied to any linear operation across a neural network -- not only FFNs/MLPs, but also convolutions, attention computations, feature projections and other components**.
>     -  Importantly, these approaches are not mutually exclusive—in fact, **our experimental setups already include gated modules** (i.e., GELUs in ViTs and GPT-2s, and SwiGLUs in LLaMAs).
>
> 2. Comparison with a more appropriate baseline
> - A more suitable baseline with similar quadratic interaction reformulates linear layers as: $z = (x @ W_1)*(x@W_2) + x@W_3$. We have included new experiments comparing QuadEnhancer (QE) with this baseline (QN) using comparable model sizes. The results, summarized below, show that QE significantly outperforms QN across all datasets in general:
>
>
> |            | params(M) | imagenet1k | caltech | cifar10 | cifar100 | flowers |  food  |   pet  |   avg  |
> |:----------:|:---------:|:----------:|:-------:|:-------:|:--------:|:-------:|:------:|:------:|:------:|
> | ViT-M + QN |   5.47    |   72.15    |  92.14  |  97.79  |  85.36   |  75.39  | 86.83  | 94.05  | 86.24  |
> | ViT-M + QE |   5.40    |   75.15    |  94.03  |  98.03  |  86.88   |  87.25  | 88.59  | 94.95  | 89.27  |
> | ViT-T + QN |   2.53    |   61.17    |  85.36  |  95.81  |  79.08   |  34.50  | 81.58  | 98.31  | 76.54  |
> | ViT-T + QE |   2.47    |   65.30    |  90.32  |  97.09  |  82.59   |  75.58  | 84.63  | 91.88  | 83.91  |
>
> We hope this clarifies the positioning of QuadEnhancer in relation to gated MLPs and highlights its broader applicability and effectiveness.
>
>
> > #### W2.1: Hard to assess how meaningful the improvement is: The field is full of small changes to MLP layers, and it's usually possible to show improvement by carefully choosing tasks, hyperparameters, or model sizes.
>
> Thank you for raising this important point. We fully understand the concern about the potential influence of selective experimental setups. To address this:
> 1. Choice of Tasks: We intentionally selected well-established and widely-used benchmarks (though not the latest) in both image classification and language tasks to ensure that our evaluation is grounded and meaningful.
> 2. Hyperparameters: To maintain fairness, we directly adopted the hyperparameter configurations from prior work [1,2] for the image classification and LLM fintuning tasks, without additional tuning.
> 3. Model Sizes: Due to limited computational resources, our experiments were conducted with relatively small models. Even though, for LLaMA fine-tuning, the model sizes are still quiet standard and consistent with prior studies.
>
> > #### W2.2 The paper does not provide state-of-the-art results on any benchmark.
>
> Thank you for your thoughtful feedback. We would like to clarify the intended contribution and our perspective on benchmarking:
>
> 1. Focus on foundational improvements: Our primary goal is not to achieve SOTA results on specific benchmarks, but rather to propose a more powerful alternative to standard linear operations. We hope this can serve as a useful building block for the broader community in designing more efficient neural networks.
> 2. SOTA involves many external factors: While SOTA results are certainly valuable, they often depend heavily on task-specific tuning, architectural tweaks, and implementation tricks. We view such results as complementary rather than essential to the core contribution of this work.
> 3. Future direction: We acknowledge the value of SOTA performance and plan to explore this direction in future work. We appreciate your perspective and will keep it in mind as we continue to develop and evaluate this method.
>
> > #### W3&Q2: No analysis of scaling behavior: Architectural changes often show benefits on small models but lose their impact as model size increases. The experiments in the paper use relatively small-scale models and do not demonstrate behavior on wider range of scales.
>
> Thank you for your thoughtful comment. We have added an experiment that examines the effect of model size by varying the hidden dimension of ViT models trained from scratch on the Caltech dataset:
>
> |        |   96  |  144  |  192  |  288  |  300  |
> |--------|:-----:|:-----:|:-----:|:-----:|:-----:|
> | ViT    | 58.20 | 59.87 | 60.50 | 61.13 | 55.35 |
> | ViT+QE | 59.35 | 60.83 | 61.39 | 61.57 | 55.53 |
> | Gain   | 1.15  | 0.96  | 0.89  | 0.44  | 0.18  |
>
> From the results, we observe two key trends:
> - (i) As the hidden dimension increases, both ViT and ViT+QE initially improve in accuracy, followed by a performance drop at the largest size due to overfitting.
> - (ii) The gain from QE decreases gradually with model size. This is an expected and reasonable trend, as larger models already possess greater capacity to model complex relationships.
>
>
> > #### Q1: Am I correct that the ViT architecture used in your experiments has a two-layer MLP with GELU as the nonlinearity, and does not use SwiGLU or GEGLU? If so, why did you choose this configuration, given that GELU-based MLPs are used in older models, and most modern architectures—including recent LLMs—use SwiGLU instead?
>
> Yes, you are correct. ViTs used in our experiments employ GELUs. We chose this configuration to ensure fair comparison by strictly following the original settings from prior work [1], which used GELU-based MLPs. The same principle applies to our LLaMA fine-tuning experiments, where SwiGLU is part of the original architecture and thus remains unchanged.
> For further discussion on gated MLP variants, please refer to our response to W1.
>
> > #### Q3: Could you clarify how exactly you combined your method with LoRA? If the base architecture remains unchanged (e.g., still uses a two-layer GELU MLP), how does the quadratic enhancement interact with the low-rank LoRA update? Did you modify the A @ B structure to insert a quadratic term between them?
>
>
> The linear transformation in LoRA-based fine-tuning is expressed as: $y = (W_0 + AB)x$. We treat the combined matrix as a standard weight matrix $W=W_0+AB$, reducing it to a standard linear form $\\tilde{y}=Wx$. This aligns with the formulation in equation (1) of our manuscript, allowing us to apply QuadEnhancer directly on top of the LoRA-adapted weights without altering the LoRA structure itself. The final output after applying QuadEnhancer becomes:
> $\\begin{align}\\tilde{y}&=(W_0+AB)x\\\\z&=(\\Lambda \\tilde{y}) \\odot \\tilde{y}+\\tilde{y}\\end{align}$.
> Thank you for pointing this out. We have added this clarification in the revised version of the manuscript.
>
>
> ---
>
>
> [1] Mehta, Sachin, Farzad Abdolhosseini, and Mohammad Rastegari. "Cvnets: High performance library for computer vision." Proceedings of the 30th ACM International Conference on Multimedia. 2022.
> [2] Hu, Zhiqiang, et al. "Llm-adapters: An adapter family for parameter-efficient fine-tuning of large language models." arXiv preprint arXiv:2304.01933 (2023).

---

> > ### Comment · Reviewer_gte1 · 2025-08-06
> >
> > Thank you for the detailed response. I appreciate the authors' detailed and thoughtful response, as well as the effort to include new experiments. All of my questions have been addressed, and the paper is now much clearer to me.
> >
> > However, I remain unconvinced by the authors' clarifications and would like to elaborate on my concerns:
> >
> > 1. I do not agree with the argument that gated linear units are merely “complementary” to the proposed method and thus irrelevant for comparison. Standard nonlinearities are pointwise: they take input tensors of shape [..., N] and output tensors of the same shape, with no interaction across dimensions. In contrast, gated linear units such as SwiGLU produce output tensors of shape [..., N/2], where each element is the result of interaction between pairs of transformed features. Similarly, the proposed quadratic enhancer produces [..., N] outputs, with each output depending on interactions among `k` neighboring elements (e.g., 3 in the reported experiments). While it is true that one could combine these techniques, the same is true for countless other function compositions. This does not justify ignoring such a strong and widely adopted baseline—especially given that SwiGLU is the default in most modern LLMs.
> >
> > 2. Regarding the authors’ statement: “The gain from QE decreases gradually with model size. This is an expected and reasonable trend, as larger models already possess greater capacity to model complex relationships” — I do not agree that this trend is expected. In fact, I would expect a more expressive nonlinearity to provide greater benefits when paired with a more expressive backbone. A decreasing gain may instead indicate either a weak or redundant modification, or that the benchmark dataset is too saturated to meaningfully reveal differences, making it an inappropriate setting to evaluate the method’s true impact.
> >
> > 3. More broadly, I struggle to distinguish this work from a long line of similar papers proposing minor modifications to MLPs or activation functions. While such methods often report modest improvements, they rarely have lasting impact or widespread adoption. I do not see clear evidence that this work breaks that pattern.
> >
> > In summary, I find the provided evidence insufficient to change my assessment, which remains unchanged.

---

> > > ### Author Response · Authors · 2025-08-08
> > >
> > > Thank you very much for your thoughtful and constructive feedback. We appreciate your careful reading and the opportunity to clarify our earlier response.
> > >
> > >
> > > > ### 1. Regarding the SwiGLU Baseline:
> > >
> > >
> > > In our initial rebuttal, we argued that SwiGLU is not a direct baseline because it is typically used within MLPs and coupled with specific activation functions, while our proposed method (QE) is designed to be a general-purpose enhancer applicable to a broader range of operations—not just MLPs.
> > >
> > > However, we now realize that your perspective is valid: if we treat SwiGLU as a nonlinear transformation rather than strictly as an "activation function," then it indeed serves as a strong and directly relevant baseline. We apologize for our earlier misunderstanding. In response, we have conducted **a new experiment comparing QE directly against SwiGLU**, with matched parameter counts achieved by adjusting hidden dimensions.
> > >
> > > We evaluated the following methods:
> > >
> > > - QN: $z = (x @ W_1)*(x @ W_2) + x @ W_3$
> > > - SwiGLU: $z = (x @ W_1) * \text{sigmoid}(x @ W_1) * (x @ W_2)$
> > > - QE (our method)
> > >
> > > The accuracy ($\\uparrow$) results are as follows.
> > >
> > > |              | parameter(M) | imagenet1k | cifar10 | cifar100 | food   | avg    |
> > > |--------------|--------------|------------|---------|----------|--------|--------|
> > > | ViT-T+QN     | 2.53         | 61.17      | 95.81   | 79.08    | 81.58  | 79.41  |
> > > | ViT-T+SwiGLU | 2.58         | 63.25      | 96.76   | 80.58    | 83.91  | 81.13  |
> > > | ViT-T+QE     | 2.47         | **65.30**      | **97.09**   | **82.59**    | **84.63**  | **82.40**  |
> > >
> > >
> > > These results confirm that SwiGLU outperforms the QN, while QE consistently surpasses both.
> > >
> > > Your comment also helped us gain a better understanding of why QE may offer superior performance. SwiGLU increases cross-dimensional interactions at the cost of reduced output dimensionality, whereas QE preserves the original dimensionality while introducing localized interactions. This design may allow QE to strike a favorable balance between preserving feature space and enhancing nonlinearity.
> > >
> > > > ### 2. Regarding the Scaling Behavior:
> > >
> > >
> > > You're absolutely right to question the original explanation we gave regarding scaling. Our earlier observation—that QE’s performance gains diminish with larger models—was based on experiments where model size increased but dataset size remained fixed. As a result, the models became overparameterized for the task, saturating performance and masking the potential benefits of QE.
> > >
> > > We fully agree with your assertion: "a more expressive nonlinearity should offer greater benefits when paired with a more expressive backbone", **provided that the dataset size scales accordingly**. To test this, we conducted a new experiment where we increased both model capacity (via hidden dimension $d$) and training dataset size $s$. The accuracy ($\\uparrow$) results are as follows:
> > >
> > >
> > > |        |  d=24,s=50k  |   d=48,s=100k  |   d=96,s=200k  |  d=192,s=400k  |
> > > |--------|:----:|:-----:|:-----:|:-----:|
> > > | ViT    | 8.99 | 19.18 | 33.14 | 49.59 |
> > > | ViT+QE | 9.06 | 19.95 | 34.03 | 50.78 |
> > > | Gain   | **0.07** |  **0.77** |  **0.89** |  **1.19** |
> > >
> > >
> > > **With both the model and data scaling up, we observe that QE delivers increasing gains, aligning with your expectation.**
> > >
> > >
> > > Once again, thank you for your valuable feedback. It has helped us refine both our experiments and our understanding.

---

### Note · Authors · 2025-08-12

To all ACs and reviewers:

Thank you very much for handling our submission and for providing insightful comments. We sincerely appreciate your time and effort in the review process.

---

To Reviewer gte1:

We would like to kindly remind you that **we have submitted a follow-up comment below your review, addressing your concerns**:

1. Lack of baseline SwiGLU: We have added comparative experiments including SwiGLU.

2. Unexpected scaling behavior: We have explained the reason for this observation and added new results that align perfectly with your expectations.

We would greatly appreciate it if you could take these additional results into account in your final justification.

---

### Decision · Program_Chairs · 2025-09-17

**Decision:**

Accept (poster)

**Comment:**

On this work, the authors introduce QuadEnhancer, a novel idea that uses quadratic transformations in neural networks to further increase the nonlinearity of the model. To do so, the authors introduce several technical innovations: Low-rank matrices for the quadratic terms, weight sharing, and sparsification of the weight matrix, with the idea being not only improving the performance but keeping the flops of the model not explode.

The paper initially received mixed scores: 2 Borderline Rejects, 1 Borderline Accept and 1 Accept. The reviewers were pretty much unanimous that the paper is well-written and makes sense, and that the results show improvement in several benchmarks. However, there were some issues with the experimental setup (the method is neither SOTA, nor used in particularly large networks) and very importantly some conceptual issues (gated activations can be considered as quadratic transformations). The authors provided a rebuttal showing more experiments and explanations to the reviewers.

After the rebuttal, two of the reviewers kept their scores (Borderline Reject and Borderline Accept), 1 reviewer increased their score from Borderline Reject to Borderline Accept, and one reviewer decreased their score from Accept to Borderline Accept, effectively having a stalemate of the scores. Thus, I carefully read the paper, the rebuttal, the reviewers-authors exchange and the internal discussion. Afterwards, summarizing my thoughts:

- The idea of the paper makes sense, and the authors have done a decent job in keeping the number of parameters and flops from exploding.

- The paper is very well-written, it's hypothesis are supported by a good experimental setup, and the paper provides code which is appreciated.

- Unfortunately, I remain unconvinced when it comes to how the method would work in complex tasks of really large neural networks, especially when combined with gated non-linearities which somehow, albeit orthogonally, tackle the same problem.

The AC and the SAC had a discussion to reach a decision for the paper. They agreed that the paper has merits, with the idea being intuitive and well-explained and the results somehow validating the idea. While they are skeptical if the method would work on larger models, on balance of things, they agree that the paper is more on the accept regime and recommend it to get accepted. Congratulations to the authors and please ensure to integrate all the results from the rebuttal in the paper!